# Roles of Physicochemical and Structural Properties of RNA-Binding Proteins in Predicting the Activities of Trans-Acting Splicing Factors with Machine Learning

**DOI:** 10.3390/ijms23084426

**Published:** 2022-04-17

**Authors:** Lin Zhu, Wenjin Li

**Affiliations:** Institute for Advanced Study, Shenzhen University, Shenzhen 518060, China; 2060391003@email.szu.edu.cn

**Keywords:** machine learning, partial least square regression, minimum redundancy–maximum relevance, forward searching strategy, hydrophobicity, secondary structure, amino acid compositions

## Abstract

Trans-acting splicing factors play a pivotal role in modulating alternative splicing by specifically binding to cis-elements in pre-mRNAs. There are approximately 1500 RNA-binding proteins (RBPs) in the human genome, but the activities of these RBPs in alternative splicing are unknown. Since determining RBP activities through experimental methods is expensive and time consuming, the development of an efficient computational method for predicting the activities of RBPs in alternative splicing from their sequences is of great practical importance. Recently, a machine learning model for predicting the activities of splicing factors was built based on features of single and dual amino acid compositions. Here, we explored the role of physicochemical and structural properties in predicting their activities in alternative splicing using machine learning approaches and found that the prediction performance is significantly improved by including these properties. By combining the minimum redundancy–maximum relevance (mRMR) method and forward feature searching strategy, a promising feature subset with 24 features was obtained to predict the activities of RBPs. The feature subset consists of 16 dual amino acid compositions, 5 physicochemical features, and 3 structural features. The physicochemical and structural properties were as important as the sequence composition features for an accurate prediction of the activities of splicing factors. The hydrophobicity and distribution of coil are suggested to be the key physicochemical and structural features, respectively.

## 1. Introduction

Alternative splicing (AS) is one of the major contributors to the functional complexity of the human genome and occurs in ≥90% of human genes [1,2]. Abnormal AS, such as mutations in cis-acting sequence elements in pre-mRNA and trans-acting splicing factors, are the common causes of a large number of human diseases [3]. AS is regulated by splicing factors that specifically bind to cis-elements in pre-mRNA to modulate the recognition of splice sites nearby [4,5]. The same splicing factor can promote or inhibit the inclusion of an exon in the spliced product depending on the location of the same cis-element that it binds to in pre-mRNA [6]. Recognizing the activities of splicing factors in developmental or differentiation contexts is essential for a better understanding of heart [7], brain [8], and liver development [9], and T-cell activation [10]. During the past decade, many studies have been performed to identify the splicing targets of individual splicing factors and their binding sites in developmental contexts, characterize the activities of the splicing factors in splicing coordination, and then infer the potential functions of these splicing networks for tissue and organ development [11,12]. For example, muscleblind-like proteins have been introduced to induce embryonic splicing patterns for more than half of the developmentally regulated AS transitions [13]. Polypyrimidine tract-binding protein 2 promotes neuronal development and tissue maintenance [14], and splicing-regulatory (SR) proteins promote exon inclusion by specifically binding to their cognate targets within an alternative exon [15]. Additionally, machine learning models have been successfully applied to explore the regulation of AS outcomes by using genomic data in the era of big data [12,16,17,18], specifically to identify gene targets of a therapeutic for human splicing disorders [12], predict the RNA polymerase II pausing events from the contextual DNA sequences [17], and identify effective exon skipping using antisense oligonucleotides [18].

Splicing factors contain both functional domains that directly participate in splicing and RNA-binding domains that bind specifically to the cis-elements in pre-mRNA. Thus, splicing factors are RNA-binding proteins (RBPs). According to a recent proteomic analysis, there are approximately 1500 RBPs in the human genome [19]. It is not clear how many of these RBPs are splicing factors. These RBPs are likely to regulate splicing because most of them are located in the nucleus. Experimentally measuring the activity of each RBP by engineering splicing factors [20] is impractical, motivating the development of predictive models for understanding the activities of these RBPs. Computational studies to identify splicing-regulatory cis-elements have been frequently reported [21,22,23,24,25,26,27,28]. Fairbrother et al. identified ten motifs with exonic splicing enhancer activity by analyzing exon–intron and splice site composition in human genes [21]. Xiong et al. trained a deep neural network on exon skipping events and constructed a promising model to predict the exon inclusion probability [22]. Recently, a machine learning model of alternative splicing, which enables the identification of universal rules of RNA splicing, was learned from gene libraries with millions of synthetic sequences [23]. However, the identification of potential trans-acting splicing factors via computational approaches is still limited. To unveil the activities of these RBPs in splicing, Mao et al. accessed the splicing activities of 63 putative functional domains from 51 human RBPs by engineering them into an RNA-binding domain with programmable specificity. They found that more than 80% of these domains possess nontrivial activities to regulate splicing. In addition, they constructed a model using a machine learning approach to predict the splicing-regulatory activities of RBPs from their sequence compositions [1]. However, they took only the amino acid composition and dipeptide composition as features to represent the RBPs. In this study, we hypothesize that a model of splicing factor activity that includes physicochemical and structural features can outperform the model utilizing only sequence composition features. Physicochemical properties have recently been used to improve the performance of machine learning models for the evaluation of target diversity and compound promiscuity in protein–drug interactions [29], identification of 5-methylcytosine sites [30] or N6-methyladenosine sites [31] in RNA sequences, classification of protein domains with highly variable sequences [32], and characterization of intrinsic properties of protein–protein interfaces [33]. Xu et al. incorporated physicochemical properties in a deep sparse autoencoder to identify splicing sites and demonstrated the constructed predictor (named iSS-PC) to be superior to other predictors for the same purpose [34]. Structural information is fundamental for understanding protein functions because more than 188,000 biological macromolecular structures are present in the protein data bank at the atomic or near atomic level. For example, the complex structure of an RNA recognition motif of the splicing factor SRSF1 with RNA was recently determined. The structure revealed a bimodal mode of interaction of SRSF1 with RNA and deepened the understanding of how SRSF1 activates the inclusion of the SMN1 (survival of motor neuron 1) exon7 [35]. However, the structures for most of RBPs are not available. Thus, the inclusion of structural information of RBPs can only rely on structure prediction tools. Physicochemical and structural properties were demonstrated to be essential in the prediction of protein structure classes [36,37], DNA binding protein identification [38], and other functional properties of proteins [39,40,41,42]. Thus, in this study, we explore whether the physicochemical and structural properties of RBPs play non-negligible roles in predicting the context-dependent activities of RBPs in regulating splicing. Specifically, we aim to determine which physicochemical and structural properties are the most important ones in such predictions.

In this study, we first included physicochemical and structural properties together with the single and dual amino acid compositions to represent RBPs. Then, minimum redundancy-maximum relevance the (mRMR) method and forward feature searching strategy were combined to determine the best feature sets and optimize the prediction model using a machine learning approach, partial least squares regression (PLSR), followed by model evaluation with 5-fold cross-validation. Finally, the roles of physicochemical and structural properties in predicting the activities of RBPs are discussed.

## 2. Results and Discussion

### 2.1. Comparing the Performance of Different Machine Learning Approaches

To choose the suitable machine learning algorithm, we compared the performance of three machine learning approaches PLSR, RFR, and SVMR. In terms of accuracy and computational time, the PLSR model was superior to the other two regression models, RFR and SVMR; 647 features were used to train the three regression models, and the PLSR model produced the highest Spearman’s correlation coefficient (γ) 0.69. The Spearman’s coefficient (γ) of RFR and SVMR were 0.53 and 0.54, respectively. Computation times for a single 5-fold cross validation on 647 features in PLSR, RF, and SVMR were 1.8 s, 330 s, and 3.6 s, respectively. These results demonstrate the superior fitting ability of PLSR. Therefore, in the following sections, only the results of PLSR are presented.

### 2.2. Determining the Parameters of PLSR for an Optimal Performance

As the performance of PLSR depends on the number of principal components used in the model, we tested the performance of PLSR using different principal components. Each protein was represented by a 647-dimensional numeric vector, and for 85 samples, an original feature matrix X (85×647) was obtained in this work. First, the feature matrix X (85×647) was trained using the PLSR model within the number of principal components from 1 to 10. The results (Figure 1, Appendix A) show that the PLSR model achieves the best performance when the number of components =3, with $R^{2}$, RMSE, NRMSE, Pearson’s correlation coefficient, and Spearman’s rank correlation coefficient (γ) of 0.47, 0.32, 0.72, 0.77, and 0.69, respectively. Therefore, we used the PLSR model with the number of components = 3 to analyze the feature importance in the following sections.

### 2.3. Feature Selection with mRMR Method

By using the mRMR feature selection program developed by Zhang [43], we obtained two feature lists (Appendix A): (1) the maxRel feature list, in which the features are listed in a decreasing relevance to the target; (2) the mRMR feature list, which is selected and ordered based on mutual information. To determine how many foremost features in the mRMR feature list should be included in the prediction model, the features in the mRMR feature list were added one by one, and the performance of the prediction model with the selected feature subset was evaluated using 5-fold cross-validation. Here, we ran all the features in the mRMR feature list and obtained the optimized feature subset that achieved the highest prediction accuracy (Spearman’s correlation coefficient of 0.71) when the 625th feature was added (Figure 2, Appendix A). The PLSR model obtained a higher Spearman’s coefficient based on the first 625 features in the mRMR feature list than that based on the 647 features. Therefore, a feature vector with a dimension smaller than the original feature vector (627-dimensional) was formed with 625 features based on the mRMR method. Similar results can be obtained when metrics such as $R^{2}$, RMSE, NRMSE, and Pearson’s correlation coefficient are used (Appendix A).

### 2.4. Feature Selection with Forward Feature Searching Strategy

Using the mRMR method, a feature subset with 625 features was obtained. However, the 625-dimensional feature vector is still a high-dimensional feature vector. Therefore, in this section, we attempt to find a better feature subset than the feature subset obtained using the mRMR method. The mRMR feature selection is fulfilled without involving the PLSR prediction model. The mRMR method can obtain the feature selection order quickly, whereas the forward feature searching strategy can provide a more accurate and better feature subset, which involves a prediction model.

This study combined the mRMR method with the forward feature searching strategy to obtain a better feature subset. We chose the first several features from the list of mRMR features and the remaining features by using the forward feature searching strategy. Our method aims to find a feature subset with few features that have the highest Spearman’s coefficient. The reasons to use Spearman’s coefficient to evaluate the model performance are as follows: (1) five metrics are correlated (Appendix A); (2) predicting the correct rank of the activities of RBPs is crucial for the selection of potential inhibitors or activators.

We chose the first two features in the mRMR feature list as the initial feature subset because the PLSR model requires at least three features when the number of principal components is 3. We then searched the remaining features using the forward feature searching strategy, a feature order table and a performance assessment table (Appendix A). Figure 2 shows that the best feature subset is between 20 and 100 features in the forward feature order list. The highest prediction Spearman’s coefficient was 0.94 when 93 features (6 from physicochemical properties, 17 from structural information, and 70 from sequence compositions) were selected in the feature subset to compare the fitted values and experimentally measured values (Appendix A). When more than 93 features are selected, the Spearman’s coefficient deteriorates, indicating that the PLSR model suffers from the overfitting problem. Therefore, the 93-feature subset is regarded as the optimal feature subset in this work.

### 2.5. Comparison between mRMR Method and Forward Feature Searching Strategy

In this study, we explore two questions: What is the difference between the feature list ordered by the mRMR method and the forward feature searching strategy? Which feature extraction method is more efficient? To answer these questions, we divided the 647 features into 13 subsets according to the order list. Their prediction accuracy (Spearman’s coefficient) is shown in Figure 3, where the Spearman’s correlation coefficients of the subsets in mRMR are not significantly different. The first subset in the forward feature searching strategy achieved the highest value. Overall, the forward feature searching strategy achieved a more accurate prediction. Therefore, the forward feature searching strategy should be first considered when the computational resources are sufficient.

### 2.6. Feature Importance Analysis

After obtaining the best feature subset with 93 features, we evaluated the relative contributions of the features obtained from physicochemical properties, structural information, and sequence compositions. Thus, 6 physicochemical property features, 17 structural information features, and 70 sequence composition features in the best feature subset were separated into three new feature subsets. For each feature subset from a particular property, a prediction model was constructed based on PLSR, and its performance was evaluated (Figure 4). The Spearman’s coefficients for the physicochemical subset (6 physicochemical property features), the structural subset (17 structural information features), and the sequence subset (70 sequence composition features) were 0.61, 0.30, and 0.59, respectively. The prediction ability of 6 physicochemical property features is comparable to that of 70 sequence composition features. Thus, the physicochemical properties are very important in the function of splicing factors. Although the sequence composition is as important as the physicochemical properties, structural information is indispensable in accurately predicting RBPs’ splicing activities.

Figure 2 shows the first 24 features of the forward searching features achieve good performance, and the Spearman’s coefficient is 0.92. Table 1 shows the 24 foremost features obtained using the forward feature researching strategy. The first 2 features (backbone_angles_ϕ_3th and hydrophobicity_distribution_H-0.0) are taken from the mRMR features in the forward searching feature list. Here, we hypothesize that the physicochemical and structural properties contribute significantly to the prediction of the activities of RBPs. To verify our hypothesis, we further analyzed the contributions of features as follows. The foremost 24 features consist of 16 Dual-AAC, 3 physicochemical, and 5 structural features. The number of Dual-AAC features dominates in the top 24 features, highlighting the importance of Dual-AAC in predicting the activities of splicing factors. This is consistent with the results of a previous study [1]. Although the occurring frequencies of several dual-features such as SR and RS are very high in splicing factors, they are not selected in the top 24 list because the values of these features do not change significantly, and thus the prediction model is not sensitive to them. However, the top 24 list contains many physicochemical and structural features as well, in which the features related to hydrophobicity and coil structure are the majority. Hydrophobicity is the driving force for protein folding and plays an important role in protein–protein interactions [44,45,46]. Our results suggest that hydrophobicity interactions can be dominant in the splicing complex formed between RBPs and other proteins involved in splicing. Furthermore, the appearance of two structural information features, SS8_distribution_C-1.0 and SS3_distribution_C-1.0, in the top 24 list indicates that the coil structures tend to be at the C-terminal of RBPs, and these low-complexity domains in the C-terminal can be important for the function of splicing factors (note that functional domains of splicing factors are mainly low-complexity domains). We showed two SHAP plots in the Appendix A) for feature importance analysis. These SHAP plots are shown based on the 5-fold cross-validation. As a powerful method for selecting features and analyzing feature importance, these SHAP plots show similar results to the combined mRMR and forward feature searching strategy. The top10 features of the list of mean (|SHAP|) values and the list of mRMR and forward feature searching strategy are the same. The SS8_Dual_GG feature in the list of SHAP has a more significant contribution to predicting the activities of RBPs. This is consistent with the result that structure information and physiochemical properties are critical in predicting splicing factors.

### 2.7. Comparison with the Existing Model

In the only existing work for predicting the activities of trans-acting splicing factors based on machine learning, 123 features (consisting of both AAC and Dual-AAC) were selected to predict the activities of RBPs based on the PLS model developed by Wang’s group [1]. Thus, we compared the performance of the 123 features from Wang’s group and the performance of 93 features (the best feature subset) obtained in this work. Here, we trained the PLSR model with the number of components ranging from 1 to 10 using the 123 features in Wang’s work based on 85 RBP samples, which are also used in this work. The highest $R^{2}$ (0.41) and Spearman’s coefficient values (0.65) were obtained using the PLSR method when the number of components was 1 (Appendix A). When the number of components was 3, $R^{2}$ and Spearman’s correlation coefficients were 0.35 and 0.58, respectively. As shown in Figure 5, both the $R^{2}$ (0.60) and the Spearman’s coefficient (0.94) achieved in this work using 93 features are higher than those achieved using 123 features in Wang’s work. Thus, in this study, a machine learning model with better performance than that of the previous work was obtained. These results highlight the need to include the physicochemical properties and structural information of RBPs in prediction models.

## 3. Materials and Methods

The overall workflow for constructing the prediction model involved five major stages: Data set collection, feature extraction, feature selection, and model training and validation (Figure 6).

### 3.1. Data Collection

We collected 91 putative functional domains for which the splicing regulatory activity in the exonic context was tested in the experiments by Mao et al., in which 63 domains and the rest were from the training and testing datasets, respectively [1]. After removing some RBPs whose structures could not be predicted by SPOT-1D [47], we obtained 85 experimentally tested RBPs sequences and their activities as the training dataset in this study. Here, PLSR was used to construct predictive models to fit the value $log_{10}$ (fold change in splicing factor activity) of each RBP. The sequences and splicing regulatory activities for these functional domains are listed in Appendix A.

### 3.2. Feature Extraction

To construct models that predict the activities of RBPs based on their sequence, the amino acid sequences of these proteins need to be converted into equal-length vectors via feature extraction [48]. In this section, we presented proteins with three main types of features: sequence compositions, physicochemical properties, and structural information, which are widely used in sequence coding research.

#### 3.2.1. Sequence Composition Features

Extracting features based on distinguishable patterns of the protein sequences is the oldest known and most common method of presenting proteins and has been widely used in several studies [41,49,50,51].

Amino Acid Composition

Amino acid composition(AAC) is a vector containing 20 elements, each of which corresponds to the frequency of an amino acid type in the entire protein sequence [49], i.e., a given protein **P** is defined by a vector in a 20-dimensional space according to the following Equation (Equation 1): (1)$$P=\left[\begin{array}{c} p_{1}\\ p_{2}\\ \vdots \\ p_{20} \end{array}\right]$$
with
$$p_{i}=\frac{c_{i}}{len(seq)},\;i=1,2,\cdots ,20$$
where $c_{i}$ represents the number of occurrences of type *i* native amino acid in the protein sequence **P**, len(seq) is the length of the sequence, and $p_{i}$ represents the occurrence frequency of amino acid *i* in protein **P**.

Dual Amino Acid Composition

To better represent the protein, another sequence composition feature, dual amino acid composition (Dual-AAC), was considered in this work. Dual-AAC can translate a sequence of protein into a 400 dimensional numerical vector, which can be described using the following Formula (Equation 2): (2)$$D=\left[\begin{array}{c} D_{1,1}\\ D_{1,2}\\ \vdots \\ D_{20,19}\\ D_{20,20} \end{array}\right]$$
with
$$D_{i,j}=\frac{f_{i,j}}{\sum_{i=1}^{i=20}\sum_{j=1}^{j=20}f_{i,j}},\;i=1,2,\cdots ,20;\;j=1,2,\cdots ,20$$
where $f_{i,j}$ is the number of transitions from type *i* native amino acid to type *j* native amino acid in a whole protein sequence and $D_{i,j}$ is defined as the changing frequency of two *i*, *j* native amino acids. In the sequence composition, Dual-AAC can be considered the composition of K-spaced amino acid pairs (CKSAAPs) with K =0 [52]. CKSAAPs was successfully employed in predicting the potential palmitoylation sites [51] and human ubiquitination sites [50].

#### 3.2.2. Physicochemical Property Features

For a given protein sequence, we used 84 physicochemical features to represent its global information. The physicochemical features include the following properties: hydrophobicity, normalized Van Der Waals volume, polarity, and polarizability; a global description of the amino acid sequence can be used to obtain 21 features for each of these properties [53,54,55]. For example, the procedure for obtaining the global properties for hydrophobicity is the following: first, we classify each amino acid into three categories polar, neutral, and hydrophobic amino acids, and then amino acids falling into the three categories are substituted by characters P, N, and H, respectively. In this step, the protein sequence is converted to a pseudosequence consisting of P, N, and H. Then, we calculate the composition (the occurrence frequencies of P, N, and H in the whole sequence), transition (changing frequencies between two different properties, for example, changing frequencies between P and N include both the transition from P to N and the one from N to P), and distribution (portion of the protein sequence that contains 25%, 50%, 75%, and 100% of P, N and H, respectively) [54,55]. Therefore, the hydrophobicity property of RBPs can be represented by 21 features: 3 composition features, 3 transition features, and 15 distribution features.

#### 3.2.3. Structural Descriptor

Using protein structural information to improve the prediction performance was demonstrated in a number of bioinformatical applications [52,56,57]. We thus introduced structural information of splicing factors in this study. As the functional domains of splicing factors are largely intrinsically disordered low-complexity domains, extracting the structural information from the amino acid sequences of RBPs is challenging. Recently, Zhou et al. developed a powerful tool (SPOT-1D) for protein structure prediction (https://sparks-lab.org/server/spot-1d/ (accessed on 10 February 2020)) [47]. We used SPOT1D to predict the one-dimensional (1D) structural properties of RBPs. The structural information of RBPs was extracted from the output files of SPOT-1D. Therefore, we obtained 3-state secondary structure (SS3), 8-state secondary structure (SS8), solvent accessible surface area (ASA), half-sphere exposure (HSE), backbone angles (θ, τ, ϕ, and ψ) and contact numbers (CN) for each residue in an RBP and converted them into structural features for model training.

Features Constructed from SS3 and SS8

One-dimensional structural properties of proteins include SS3 and SS8, which means 3-state and 8-state secondary structural elements, respectively. As important protein descriptors, SS3 and SS8 are known for only a relatively small number of proteins; thus, we predicted them from amino acid sequences using SPOT-1D [47]. SPOT-1D provided independent predictors for both SS3 and SS8, and SS8 is defined as $3_{10}-helix$ (G), α−helix (H), π−helix (I), β−bridge (B), β−strand (E), high−curvatureloop (S), β−turn (T), and coil (C). These can be condensed into three types (SS3): helix H (G, H, and I in SS8), strand E (B and E in SS8), and coil C (everything else). Therefore, SS8 can be viewed as pseudosequences that consist of G, H, I, B, E, S, T, and C, whereas SS3 can be viewed as pseudosequences that consist of H, E, and C. Then, the composition, dual composition and distribution of the queried pseudosequences are presented. Therefore, 27 features (3 composition features; 9 dual composition features; and 15 distribution features) are included in SS3 and 112 features (8 composition features; 64 dual composition features; and 40 distribution features) are included in SS8.

The probability of an amino acid belonging to C, E, or H in the case of SS3 is provided in the output files of SPOT-1D. For a protein **P** with *L* amino acids, three probabilities for each amino acid are given, and thus three *L*-dimensional numerical vectors are provided. We converted each of these probability vectors into three position-dependent features using the following strategy. We first equally divide an *L*-dimensional numerical vector into three parts: the first, middle, and end parts. Then, the average value in each part is used as the value of the feature for the corresponding part. Therefore, there are 3 features for each *L*-dimensional numerical vector and thus 9 features in the case of SS3. Together with the 27 features constructed previously, there are 36 features for SS3 in total. Similarly, there are eight probability vectors for the secondary structure types in the case of SS8, and 24 additional features are obtained besides the 112 features obtained previously. In total, there are 136 features for SS8. Therefore, a 172-dimensional feature vector is formed based on the structural properties of both SS3 and SS8.

Features Constructed from Other Structural Properties

The other structural properties correspond to the ASA, HSEα-up and -down, CN and backbone angles, which can be obtained from the outputs of SPOT-1D [47]. Similar to the cases of probability vectors for SS3 or SS8, we convert each of these properties for the sequence of an RBP into three position-dependent features. We thus obtain 24 (3×8) features.

Finally, all features are combined into a vector to train a model. The total number of features is 20+400+84+172+24=700 (Appendix A). Each RBP sequence is transferred into a 700-dimensional vector (Table 2).

### 3.3. Feature Normalization

To avoid having a particular feature leading the prediction, we need to normalize all features using the following Equation (Equation 3): (3)$$X^{{}^{\prime }}=\frac{X-X_{mean}}{X_{std}}$$
where *X*, $X_{mean}$, and $X_{std}$ represent the original value, the mean value of *X*, and the standard deviation of *X*, respectively. $X^{{}^{\prime }}$ is the output value of *X* after normalization.

### 3.4. Feature Selection

A 700-dimension vector was constructed based on the description in the feature extraction section. However, the problem of ”curse of dimensionality” may occur when we directly input the high-dimensional feature into the models [58]. Thus, to improve the prediction performance of the predictors and save computing resources, we need to reduce the feature space to a low-dimensional one using feature selection algorithms. Here, a feature matrix X (85×647) was obtained after removing the features that are almost zero in all samples (Appendix A). Then, we selected mRMR algorithm and forward feature searching strategy to reduce the feature matrix X further.

mRMR

As a widely used feature selection algorithm, mRMR is proposed by Peng et al. [59] to select a subset of features that minimize the redundancy of the original feature space and remove features of low relevance to the target class. This algorithm is especially useful for large-scale feature selection problems. mRMR has been employed in many research fields, including predicting protein structure classes [55,60], the prediction of protein–protein interaction [61,62], and others [52,63,64].

However, the optimization of feature selection through mRMR does not guarantee that the selected subset is also the best one for the prediction model because the mRMR method does not involve a prediction model [55]. Typically, the forward feature searching strategy is applied to the results of mRMR to improve the performance of the prediction model.

Forward Feature Searching Strategy

First, an initial feature subset *A* is selected, and then the remaining features are added one by one using the following Equation (Equation 4): (4)$$\underset{i\in \Omega_{A}}{max}P\left(A\cup \left(i\right)\right)$$
where $\Omega_{A}$, *i*, P(A∪(i)) represent the remaining feature subset, one feature in the remaining feature subset, and the prediction accuracy of the model with the feature subset A∪(i), respectively. $\underset{i\in \Omega_{A}}{max}P\left(A\cup \left(i\right)\right)$ means that finding a feature *i* from the remaining feature subset lets feature subset A∪(i) obtain the maximum value of the prediction accuracy.

### 3.5. Model Training

Partial least squares is a well-established technique in multivariate data analysis and has been applied to predict the activities of trans-acting splicing factors successfully [1]. Therefore, we used PLSR as the predictor in this study. PLSR is a method used to relate two data matrices, X and Y, using a linear multivariate model [65,66]. PLSR has the ability to analyze data with strongly correlated, noisy, and incomplete variables in both X and Y. It projects the X and Y variables to a new space and forms new $X^{{}^{\prime }}$ and $Y^{{}^{\prime }}$ variables. Then, a regression model is constructed between the $X^{{}^{\prime }}$ and $Y^{{}^{\prime }}$ variables. As one of the most classic regression models, PLSR has been widely used in analyzing biological data [67,68,69,70] such as determining the secondary structure of proteins in different environments [67], extracting gene association networks from microarray data [68], and exploring subcellular responses of prostate cancer cells to X-ray exposure [69]. The number of principal components is a key parameter that affects the performance of PLSR. The parameter space {1,2,3,4,5,6,7,8,9,10} was searched to identify the number of principal components at which the optimal performance was reached in this work.

The PLSR algorithm was compared with the random forest regression (RFR) and support vector machine regression (SVMR) algorithms. RFR and SVMR are among two of the most-used regression models, and they have been previously compared with the PLSR model in various studies [71,72,73,74]. Two key parameters that affect the performance of RFR are the number of the trees (M) and the number of features (mtry), whereas the performance of SVMR with a Gaussian radial basis kernel is affected by parameters cost (C) and gamma (Γ). We used grid search to identify the optimal values of these parameters in the parameter space as follows: M ∈{1,2,⋯,99}, mtry∈{1,2,⋯,29}, C ∈{0.01,0.1,05,1.0,5,10,15,20}, and $\Gamma \in \left\{2^{-6},2^{-5},\cdots ,2^{5},2^{6}\right\}$. The three machine learning algorithms were implemented in the machine learning library sklearn (version 0.24.2).

### 3.6. Cross-Validation

Cross-validation is one of the most common methods used to estimate the performance of a model. In this study, we used 5-fold cross-validation. We randomly split the original dataset into five equal-sized subsets. For each cross-validation test, one subset was used as the testing dataset, and the remaining four subsets formed the training dataset [75]. Therefore, each subset was used once for testing and four times for training, and testing was repeated five times [52]. The average metrics over the 5-fold models were used to evaluate the performance of the model.

### 3.7. Performance Evaluation

Five metrics, the coefficient of determination ($R^{2}$, Equation (Equation 5)), the root mean square error (RMSE, Equation (Equation 6)), the normalized root-mean-square error (NRMSE, Equation (Equation 7)), the Pearson’s correlation coefficient (ρ, Equation (Equation 8)), and the Spearman’s rank correlation coefficient (γ, Equation (Equation 9)) were used to evaluate the model performance. Those equations are defined as follows:(5)$$R^{2}=\frac{\sum_{i=1}^{N}{\left(y_{pred,i}-{\bar{y}}_{obs}\right)}^{2}}{\sum_{i=1}^{N}{\left(y_{obs,i}-{\bar{y}}_{obs}\right)}^{2}}$$
(6)$$RMSE=\sqrt{\frac{1}{N}\sum_{i=1}^{N}{\left(y_{obs,i}-y_{pred,i}\right)}^{2}}$$
(7)$$NRMSE=\frac{RMSE}{\sqrt{\frac{1}{N-1}\sum_{i=1}^{N}{\left(y_{obs,i}-{\bar{y}}_{obs}\right)}^{2}}}$$
(8)$$\rho =\frac{\sum_{i=1}^{N}\left(y_{obs,i}-\bar{y}obs\right)\left(y_{pred,i}-{\bar{y}}_{pred}\right)}{\sqrt{\sum_{i=1}^{N}{\left(y_{obs,i}-{\bar{y}}_{obs}\right)}^{2}}\sqrt{\sum_{i=1}^{N}{\left(y_{pred,i}-{\bar{y}}_{pred}\right)}^{2}}}$$
(9)$$\gamma =1-\frac{6\sum_{i=1}^{N}d_{i}^{2}}{N(N^{2}-1)}$$
where *N* is the number of domains in the validation set (N=17). $y_{obs,i}$ and $y_{pred,i}$ represent the prediction value and observation value of $log_{10}(\mathrm{fold}\;\mathrm{change}\;\mathrm{in}\;\mathrm{exon}\;\mathrm{inclusion})$, respectively. ${\bar{y}}_{obs}$ and ${\bar{y}}_{pred}$ are the mean values of $y_{obs,i}$ and $y_{pred,i}$, respectively. While the Spearman’s rank correlation coefficient γ represents the Pearson’s correlation of the ranks, $d_{i}$ is the difference between the two ranks of $y_{obs,i}$ and $y_{pred,i}$.

## 4. Conclusions

The biological significance of trans-acting splicing factors has motivated the development of computational tools to predict the splicing activities of RBPs. As a natural extension of an existing work, in which only the AAC and Dual-AAC features were employed, this study incorporated the physicochemical properties and structural information of RBPs into a machine learning model using the PLSR method. We established a more accurate machine learning model than that of the existing work [1] by considering the physicochemical and structural information. We demonstrated that physicochemical and structural properties play an equally vital role compared with the sequence compositions in the accurate prediction of the activities of RBPs in alternative splicing. By combining the mRMR method and the forward feature searching strategy, we obtained 24 features, and the Spearman’s rank correlation coefficient of the constructed prediction model was 0.92. Hydrophobicity and coil structure are were the two most important features among the proposed physicochemical and structural properties. These two features have important functional implications for splicing factors: hydrophobicity interactions are the main driving forces for RBPs to form splicing complexes with other proteins involved in alternative splicing; functional domains of RBPs that are mainly low-complexity domains have the potential preference to be located at the C-terminal of the splicing factors. This study provides a reference for understanding the splicing factors and further explores the mechanism of human gene expression. Studies analyzing the relationship between the activities of splicing factors and RBP sequences based on machine learning methods are limited, and this study makes a great contribution in this field. Predicting the activities of splicing factors using machine learning approaches saves a great deal of time and money.

Future research will improve the performance of the prediction model in the following aspects: (1) the dataset; (2) the feature extraction method; (3) the machine learning approach. Currently, the number of neutral domains (with splicing activity close to zero) in the dataset is too small compared with the number of splicing activators and splicing inhibitors. The model thus has the tendency to predict a true neutral domain to be either an activator or inhibitor. The inclusion of more neutral domains in the dataset could be important to eliminate the bias in the current prediction model. Here, only four physicochemical properties were extracted. Note that there are now 566 amino acid indices in the AAindex [76,77], and each amino acid index can be used as a feature. Thus, other properties, such as amino acid charges at different pH values, average flexibility, surface tension, solvation free energy, and conformational preferences, can also be included in the feature set to achieve potentially better representations of RBPs. Only the information of secondary structure prediction using the SPOT-1D method is included in this work. The importance of structural features in the prediction model highlights the need to obtain three-dimensional structural information for a better understanding of the potential function of these RBPs in alternative splicing. Such structural information can be obtained via experimental approaches or the state-of-art computational methods such as AlphaFold [78] or RossetaFold [79]. In this study, the number of experimentally tested RBPs was too small to explore the relationship between RBP sequences and their functions using deep learning approaches. In the future, we expect to establish a public database with a greater number of experimental works and improve the performance of the prediction model by incorporating more information about splicing factors and more data. Although we demonstrated the PLSR as a promising method in the prediction of splicing activity of RBPs, we only tested three machine learning models. It would be beneficial to evaluate the performance of other machine learning models, such as artificial neural networks [80,81,82] and boosting algorithms [83,84], in the near future.

## Figures and Tables

**Figure 1 ijms-23-04426-f001:**
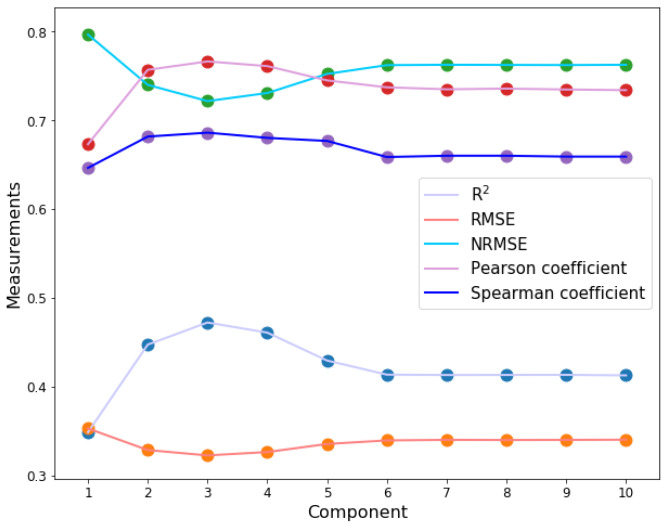
Curves of five performance measures produced by PLSR with component number ranging from 1 to 10 and 647 features trained by using the PLSR model as the input space. When the component number is 3, RMSE and NRMSE achieve the lowest value; $R^{2}$, Pearson’s correlation coefficient, and Spearman’s coefficient have the highest value. Therefore, the PLSR shows the best performance when the number of principal components is 3. $R^{2}$ is the coefficient of determination, and RMSE, NRMSE is the root mean square error and the normalized root-mean-square error, respectively.

**Figure 2 ijms-23-04426-f002:**
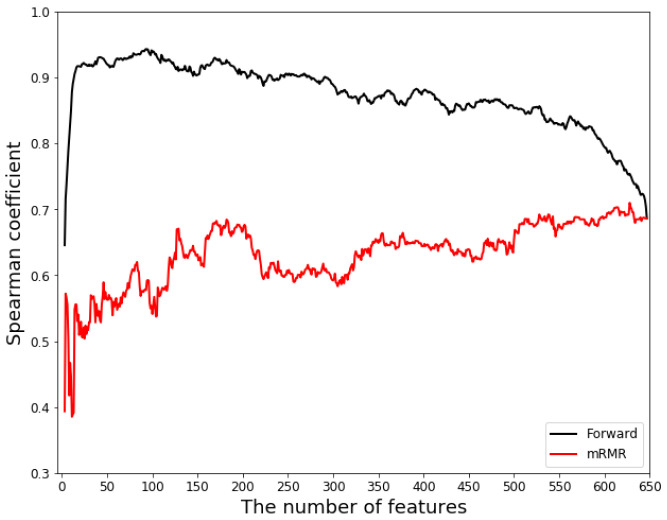
Curves of Spearman’s correlation coefficient produced by mRMR features (red line) and the forward feature searching strategy (black line). The mRMR method combined with the forward feature searching strategy has a good performance in feature selection. mRMR and Forward represents the minimum redundancy-maximum relevance method and forward feature searching strategy, respectively.

**Figure 3 ijms-23-04426-f003:**
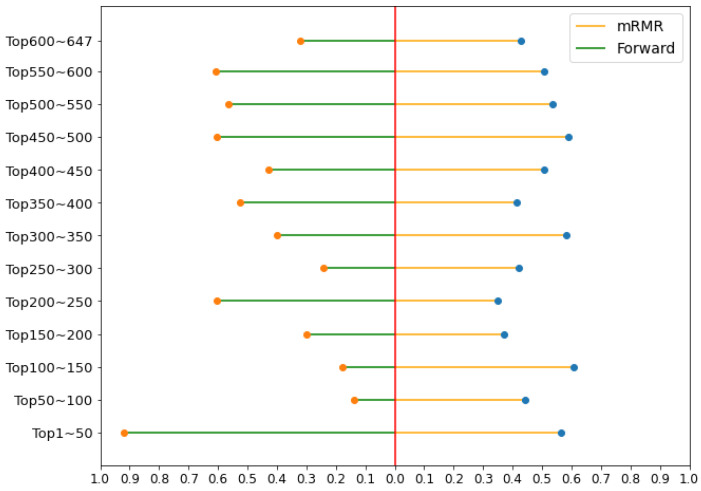
Feature importance analysis of the mRMR method and the forward feature searching strategy. The dots of the Spearman’s coefficient are produced using the mRMR features (orange lines, blue dots) and the forward feature searching strategy (green lines, orange dots). The first 50 features of the forward feature searching method achieved the highest value.

**Figure 4 ijms-23-04426-f004:**
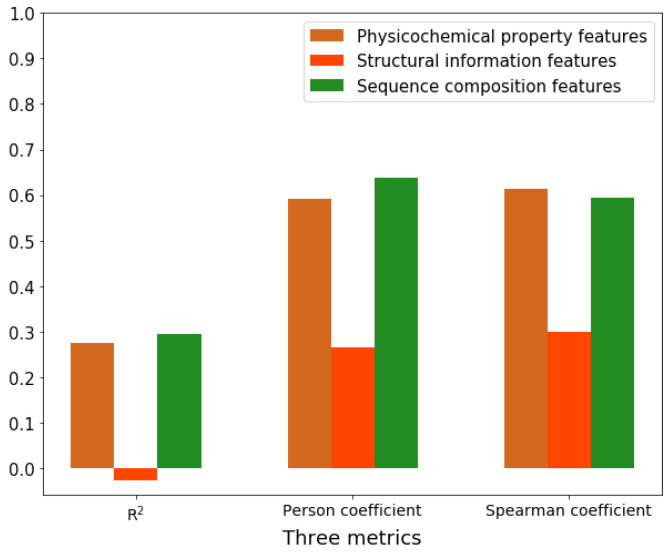
Three performance measures ($R^{2}$, Pearson’s correlation coefficient, and Spearman’s correlation coefficient) produced by 6 physicochemical features, 17 structural information features, and 70 sequence composition features based on the PLSR model when the number of component is 3.

**Figure 5 ijms-23-04426-f005:**
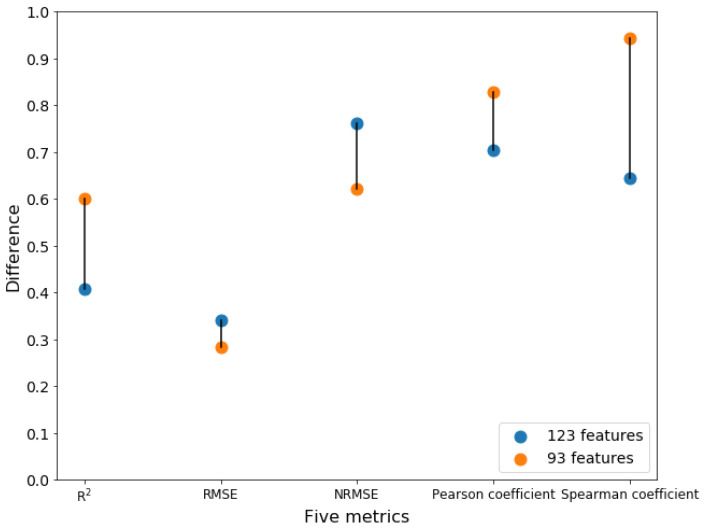
Differences in five performance measures between 93 features (the best feature subset) and 123 features (the feature subset) proposed in Wang’s work. $R^{2}$, Pearson’s, and Spearman’s correlation coefficients of 93 features are higher than those of 123 features, and RMSE and NRMSE are lower than those of 123 features.

**Figure 6 ijms-23-04426-f006:**
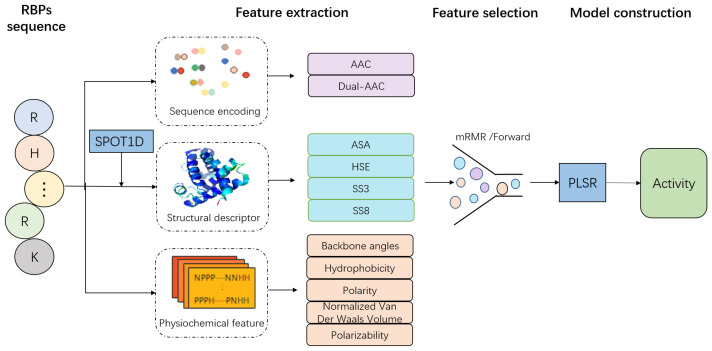
Methodology for predicting the activities of trans-acting splicing factors. RBPs is RNA binding proteins, AAC is amino acid composition, Dual-AAC is dual amino acid composition, ASA is solvent accessible surface area, HSE is half-sphere exposure, SS3 and SS8 is 3-state and 8-state secondary structural elements, respectively, and PLSR is partial least square regression.

**Table 1 ijms-23-04426-t001:** Order, names, and brief descriptions of the foremost 24 features selected by using the forward searching strategy with the performance of the corresponding prediction model measured by using the Spearman’s correlation coefficient.

Order	Name	Descriptions	Spearman
**1**	Backbone angle_tau-3th	the measurement of the residue-wise torsion	-
**2**	Hydrophobicity_distrib-ution_H-0.0	the first distribution value for H	-
**3**	Polarity_composition_P	the percentage of physiochemical property P	0.65
**4**	Dual-AAC_GT	the percentage of dual amino acid GT	0.72
**5**	Dual-AAC_HK	the percentage of dual amino acid HK	0.74
**6**	Dual-AAC_KK	the percentage of dual amino acid KK	0.76
**7**	Dual-AAC_FN	the percentage of dual amino acid FN	0.79
**8**	Dual-AAC_HY	the percentage of dual amino acid HY	0.81
**9**	Dual-AAC_TA	the percentage of dual amino acid TA	0.83
**10**	Dual-AAC_RT	the percentage of dual amino acid RT	0.85
**11**	Dual-AAC_TT	the percentage of dual amino acid TT	0.88
**12**	SS8_Dual_GG	the percentage of dual $3_{10}-helix$ G	0.89
**13**	Dual-AAC_AA	the percentage of dual amino acid AA	0.90
**14**	Dual-AAC_DP	the percentage of dual amino acid DP	0.91
**15**	Dual-AAC_IY	the percentage of dual amino acid IY	0.91
**16**	Dual-AAC_YQ	the percentage of dual amino acid YQ	0.92
**17**	Dual-AAC_VH	the percentage of dual amino acid VH	0.92
**18**	SS8_distribution_C-1.0	the fifth distribution of coil in SS8	0.92
**19**	SS3_distribution_C-1.0	the fifth distribution of coil in SS3	0.92
**20**	Hydrophobicity_distrib-ution_N-1.0	the fifth distribution of N	0.92
**21**	Dual-AAC_CG	the percentage of dual amino acid CG	0.92
**22**	Dual-AAC_CE	the percentage of dual amino acid CE	0.92
**23**	Dual-AAC_WS	the percentage of dual amino acid WS	0.92
**24**	SS8_Dual_GS	the percentage of pair $3_{10}-helix$ G and high-curvature loop S	0.92

**Table 2 ijms-23-04426-t002:** Feature description.

Feature Type	Overview of the Features
Amino acid composition	20
Dual amino acid composition	400
Hydrophobicity	21
Normalized Van Der Waal volume	21
Polarity	21
Polarizability	21
SS3	36
SS8	136
Others (ASA, HSEα—up and down, CN and backbone angles)	24
Total	700

## Data Availability

The data presented in this study are available in article and Appendix A.

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
