# Peer review of "Roles of Physicochemical and Structural Properties of RNA-Binding Proteins in Predicting the Activities of Trans-Acting Splicing Factors with Machine Learning"

_ijms, 2022, doi:10.3390/ijms23084426_

Round 1

Reviewer 1 Report

Dear Authors,

After a detailed review of the manuscript, here are a few suggestions to be incorporated to further enhance the quality of the manuscript.

  1. Kindly recheck if all the full form of the abbreviated words is written for the first time in the manuscript.
  2. Instead of writing it as Zhou and coworkers change to Zhou et al.
  3. What is the Y-axis label in Figure 2, 3, and Figure 6?
  4. Shap plots that provide the variable importance plot can be included.
  5. Authors need to justify the reason for the selection of 3 machine learning models - PLSR, RFR, and SVR in the present study
  6. A comparison of results and their differences from similar studies can be discussed with reference to the present approach in the present study to be included in the discussion section of the manuscript
  7. Authors can recheck for English language and grammatical corrections in the manuscript
  8. Authors can cite the following articles to further strengthen the work in the manuscript with the methods of data collection and machine learning models and approach https://doi.org/10.1038/s41467-021-23663-2, https://dx.doi.org/10.30919/es8d580, https://dx.doi.org/10.30919/es8d579, https://doi.org/10.1093/nar/gkab442 https://dx.doi.org/10.30919/es8d515, https://doi.org/10.1073/pnas.2007450118, 

Reviewer 2 Report

Thank you for providing me with the opportunity to read “Roles of Physico-chemical and Structural Properties in Predicting the Activities of Trans-Acting Splicing Factors with Machine Learning”. I have the following comments:

  • Please follow MDPI formatting.
  • Please get the paper proofread for language and grammar errors: Recently, an
  • attempt to build a machine learning model for predicting the activities of splicing factors is based on features consisting of single amino acid compositions and dual amino acid ones – is not clear, same goes for many other sentences.
  • Please add a couple of lines of results in the abstract. Currently, it presents the numbers for validation, such as spearman correlation but no actual results.
  • Please provide the references when you use sentences and phrases such as many studies. A single reference in such a case is not enough; the readers may want to see the “many” studies.
  • What are the implications and contributions of this study? Please highlight in detail and discuss accordingly.
  • The novelty of the study needs to be discussed in the introduction section in detail.
  • The paper needs support from the literature. Therefore, the authors should insert a separate literature section and discuss the state-of-the-art literature published on this topic.
  • How were the 85 functional domains collected? This data collection needs to be elaborated in detail.
  • Please number all the equations and refer to them in the paper. It is hard to follow these currently. Avoid using words such as following or below for equations, as any publishing change may create serious confusion.
  • The discussion section must be improved by comparison with other published papers. How is your work comparable to other similar studies? Please elaborate in detail.
  • The limitations and future directions of the study need to be further clarified in the conclusion section.
  • Also, add the practical implications of the study to the conclusion section.

ALL THE BEST !!!

Round 2

Reviewer 2 Report

Thank you for addressing my comments